# Use of Cannabis and Cannabinoids for Treatment of Cancer

**DOI:** 10.3390/cancers14205142

**Published:** 2022-10-20

**Authors:** Viktoriia Cherkasova, Bo Wang, Marta Gerasymchuk, Anna Fiselier, Olga Kovalchuk, Igor Kovalchuk

**Affiliations:** 1Department of Biological Sciences, University of Lethbridge, Lethbridge, AB T1K 3M4, Canada; 2Cumming School of Medicine, University of Calgary, Calgary, AB T2N 1N4, Canada

**Keywords:** endocannabinoid system, cancer and carcinogenesis, primary care, palliative care, cannabinol, tetrahydrocannabinol

## Abstract

**Simple Summary:**

Cancer is a disease which affects approximately 40% of people in their lifetime. Chemotherapy, the primary choice for treatment of cancer, is often ineffective or/and presents itself with many debilitating side effects, including loss of appetite, nausea, insomnia, and anxiety. Components of cannabis extracts, including cannabinoids and terpenes, may present an alternative for controlling side effects and may be used for tumor shrinkage together with chemodrugs. Cannabinoids act on so called endocannabinoid system (ECS) that operates in our body to maintain homeostasis. ECS promotes healthy development of tissues and regulates many processes in our organism and when disbalanced may lead to disease, including cancer. In this review, we will discuss the role of the ECS in relation with carcinogenesis and use of cannabis extracts and their components for primary and secondary care of cancer. Knowledge about the use of cannabinoids for cancer therapy may prolong the life of many cancer patients.

**Abstract:**

The endocannabinoid system (ECS) is an ancient homeostasis mechanism operating from embryonic stages to adulthood. It controls the growth and development of many cells and cell lineages. Dysregulation of the components of the ECS may result in uncontrolled proliferation, adhesion, invasion, inhibition of apoptosis and increased vascularization, leading to the development of various malignancies. Cancer is the disease of uncontrolled cell division. In this review, we will discuss whether the changes to the ECS are a cause or a consequence of malignization and whether different tissues react differently to changes in the ECS. We will discuss the potential use of cannabinoids for treatment of cancer, focusing on primary outcome/care—tumor shrinkage and eradication, as well as secondary outcome/palliative care—improvement of life quality, including pain, appetite, sleep, and many more factors. Finally, we will complete this review with the chapter on sex- and gender-specific differences in ECS and response to cannabinoids, and equality of the access to treatments with cannabinoids.

## 1. Introduction

Endocannabinoid system (ECS) is an ancient (over 600 mln years old), evolutionary stable animal homeostasis system [1]. It consists of three components—ligands, including 2-arachidonoylglycerol (2-AG) and arachidonoyl ethanolamide (AEA or anandamide), receptors, such as cannabinoid receptor 1 (CB1) and cannabinoid receptor 2 (CB2), and the metabolizing enzymes—fatty acid amide hydrolase (FAAH) and monoacylglycerol lipase (MAGL). As a regulator of homeostasis, ECS regulates the activity of brain, endocrine, and immune systems, among others. One such regulatory mechanism is the regulation of energy metabolism. ECS increases the energy intake, facilitates its storage, and decreases the expenditure [2]. Central regulators of energy metabolism, such as hypothalamic orexigenic neuropeptide Y (NPY) and anorexigenic cocaine and amphetamine regulated transcript (CART) peptide and peripheral regulators, such as leptin (LEP), ghrelin (GHRL) adiponectin (ADIPOQ), and cholecystokinin (CCK), are known to be dysregulated in various cancers and to contribute to malignancy (reviewed in [3]).

Genes responsible for energy homeostasis play essential roles in the organism, and dysregulation of energetic metabolism not only results in the development of metabolic syndrome, obesity, and diabetes, but is also linked to many cancers [4]. Detailed description of the role of ECS in controlling energy homeostasis is beyond the scope of this review and is presented well elsewhere [5].

Cancer is a disease of dysregulated and uncontrolled cell division and cell proliferation. Successful malignization requires mutations in multiple genes [6]. Numerous theories of cancer development and progression exist. Currently, most cancers have no cure; even so, significant progress in the development of chemotherapy and immune therapy of cancers has been achieved. Cancer therapy consists of primary care, directed at tumor eradication and palliative care, which aims to reduce side effects and suffering of a patient.

Cannabinoids as endogenous regulators of homeostasis are the molecules that can potentially be used for cancer therapy. They may be particularly useful in palliative care.

In this review, we will discuss the role of the endocannabinoid system in the control of cell growth and proliferation, describe changes in various cells and tissues that occur in ECS in response to carcinogenesis, describe major mechanism of action of endo- and phytocannabinoids on various cancers, discuss data from cell, animal, and human studies, and discuss the use of various cannabinoids for primary and palliative care.

## 2. Role of Endocannabinoids in the Human Body

ECS is active in virtually all cells of our organism. It plays an important role in the reproduction, function, and proper development of gametes [7], fertilization event, embryo implantation, and proper placenta development [8]. It is also active at all stages of embryogenesis, regulating cell division, and tissue and organ development, specifically, regulating differentiation of neural progenitors, synaptogenesis, and axonal migration [9]. During human adult life, it regulates homeostasis of many tissues, playing critical role in proper brain function by regulating neuronal synaptic communications affecting critical organismal functions, including general metabolism, growth and development, reproduction, learning and memory formation, mood, and behavior, among others [10]. In the peripheral tissues, endocannabinoids are involved in endocrine regulation and energy balance [11], as well as regulating the function of innate and adaptive immune system and immune response [12], regulating cell migration and apoptosis. The activity and functionality of ECS depends on many factors, from cell- and tissue-specific differences in the synthesis of endocannabinoids, to the number and the activity of endocannabinoid and auxiliary receptors, to the expression and the activity of enzymes involved in the degradation of circulating endocannabinoids.

In the cells, endocannabinoids acting in CB-receptor-dependent and independent manner exhibit anti-oxidative properties, are involved in clearance of damaged molecules and regulate mitochondrial activity. Anti-oxidative properties are associated with the inhibition of production of reactive oxygen species (ROS), metal chelation and prevention/alleviation of ROS-induced cell damage [13]. It should be noted that the anti-oxidative effects of cannabinoids are cell specific—while in most cells of the body, they mitigate oxidative stress, in hepatic cells they may cause it, leading to cell death [14]. Similarly, in cancer cells, such as gliomas and leukemia, cannabinoids promote oxidative stress [13].

Cannabinoids contribute to recycling of damaged molecules and are likely involved in autophagy in health tissues [15]—the activity well documented in cancer cells (discussed below). In normal cells, they increase lysosomal stability and integrity [15] through CB1 receptors found on the surface of lysosomes.

CB1 receptors are also present on the surface of mitochondria. They regulate mitochondrial oxidative phosphorylation in a positive and a negative manner, acting through the CB1 receptor, but it is not clear what modulates this activity [13]. When cells are stressed, cannabinoids attenuate mitochondrial damage [16] and decrease calcium-induced cytochrome c release [17].

### 2.1. Mechanism of Action—Ligand/Receptor

Cannabinoid receptors are ubiquitous and expressed on the cell surface as well on cell organelles, including mitochondria and lysosomes. Classical cannabinoid receptors include CB1 and CB2. CB1 is expressed at a higher level in central and peripheral nervous systems, while CB2 is expressed in many different tissues, including the immune system, internal organs, skin, bone, muscle, and glia in the brain [18]. CB1 and CB2 are GPCR (Gi/o) protein-coupled receptors, and when activated, they modulate various cellular functions through receptor internalization; interaction with other G-protein-coupled receptors; inhibition of adenylyl cyclase activity, changing the activity of calcium and potassium channels; increasing phosphorylation of various mitogen-activated protein kinases (MAPK); and many more functions [12] (Figure 1).

Endo- and phytocannabinoids interact with other receptors throughout the body, including the ionotropic transient receptor potential (TRP) cation channels family, including TRPA1, TRPV2, TRPV3, and TRPV4; nuclear receptors/transcription factors called the peroxisome proliferator-activated receptor (PPAR) α and γ; along with the orphan GPCRs, including GPR18 and GPR55; serotonin 1A receptor (5-HT1A); and the adenosine A2A receptor [19,20,21]. The nature of interaction is not always apparent, but it was shown that phytocannabinoid delta-9-tetrahydrocannabinol (THC) functions as an agonist of GPR55, GPR18, PPARγ receptors, while acting as an antagonist on TRPM8 and 5-HT3A receptors. In contrast, cannabidiol (CBD) has a very weak affinity for CB2 or CB1, although it may work as a negative allosteric regulator of these receptors [22], modulating THC activity. TRPA1, TRPV1, TRPV2, TRPV3, PPARγ, 5-HT1A, A2 and A1 adenosine receptors, and CBD functions as an agonist, while on GPR55, GPR18, and 5-HT3A, it functions as an antagonist. In addition, CBD can have inverse agonist activity on the GPR3, GPR6, and GPR12 receptors [23]. THC and CBD also can affect the levels of anandamide in the brain. Moreover, THC can increase AEA and adenosine levels [24].

### 2.2. Role in the Control of Cell Division and Cell Proliferation

It appears that ECS controls the fate of many cells in the organism, regulating the cell division and proliferation, apoptosis, necrosis and autophagy in several organs and organ systems, including the brain, skin, and immune system.

In the central nervous system (CNS), the ECS system functions as a neuroprotective system that controls glutamate excitotoxicity, calcium influx, inflammation, and autophagy [25]. In the CNS, the interaction of endocannabinoids with CB1/CB2 and other receptors mediates synaptic plasticity or progenitor cell fate in the central nervous system, promoting self-repair of the brain [26]. It also appears that constitutive release of 2-arachidonoylglycero by late oligodendrocyte progenitors allows oligodendrocyte maturation by activating CB receptors and downstream ERK pathway [27].

In skin, ECS activity maintains the cutaneous homeostasis through the regulation of skin cell proliferation, survival, and differentiation [28]. Locally produced AEA inhibits the cellular growth and the differentiation of cultured NHEK and HaCaT keratinocytes, as well as inducing apoptosis of human HaCaT keratinocytes [28]. CB1 activity is higher in differentiated skin layers [29]. In human cultured hair follicles, AEA but not 2-AG inhibit elongation and proliferation of hair shaft and induce intraepithelial apoptosis in a CB1-dependent manner [30]. Both AEA and 2-AG induce apoptosis of human sebaceous-gland-derived SZ95 sebocytes in a CB2-dependent manner [31].

In the immune system, the central role is played by CB2 receptors that are mainly expressed by cells (T and B lymphocytes) and peripheral tissues of the immune system (spleen and thymus) where it regulates immune suppression, apoptosis, and cell migration [32]. In in vitro studies, it was demonstrated that anandamide inhibits mitogen-induced proliferation of T cells [33], while inhibiting the chemokine SDF-1-induced migration of CD8+ T cells [34]. In contrast, 2-AG, but not anandamide, induced CB2-dependent migration in natural killer cell line KHYG-1 cells [35]. In B cells, 2-AG chemo-attracts naïve B cells and marginal zone B cells and inhibits the function of activated B cells, while 2-AG and anandamide suppress the migration of neutrophils [36]. Additionally, anandamide induces the apoptosis of murine bone-marrow-derived DCs (BMDCs) in a CB1- and CB2-dependent manner [37].

### 2.3. Changes in the ECS with Age

Cancer can be considered an age-associated disease, due to the accumulation of cellular and DNA damage. From this perspective, it is interesting to understand what happens to ECS with age.

In general, information about age-related changes in the ECS is scarce. Most of the data are related to changes in the central nervous system, and even then, the data are very contradictory. In general, it is believed that the activity of ECS declines with age [13]. In rats, in one study, a general decrease in the expression of CB1 and a decrease in density of the receptors in various brain areas with age was observed [38], while in another study—in which only redistribution of the receptors was noted– they were reduced in the postrhinal, but elevated in the entorhinal and temporal cortices in old animals [39] (Table 1). In mice, no changes in the receptor density in most brain regions was found with age, but instead, a significantly reduced receptor/Gi protein coupling was observed [40]. In one study on humans, CB1 expression increased, predominantly in females, most drastically in the basal ganglia, the lateral temporal cortex, and in the hippocampus [41], while another study reported no change [39]. As for endocannabinoids, the picture is not clear either—some studies suggested a decrease, while others found no difference in different brain regions of young and old animals. [13]. However, animals lacking FAAH—the enzyme degrading anandamide showed less pronounced features of aging—decreased expression of pro-inflammatory genes and decreased decline in cardiac function [42] (Table 1).

Very little reliable data exist on changes in the ECS in skin. Concentration of AEA is 119-fold higher than 2-AG in human skin [43], although it is not known how it changes with age. Moderate CB1 activation in skin works as a suppressor of the differentiation, while high activation leads to anti-proliferative and pro-apoptotic events [19]. In mice, CB1 deficiency in skin may lead to premature aging [44]. CB1-deficient mice exhibit cognitive impairments and changes in the structure of skin, indicating that CB1 deficiency accelerates aging only in the brain and in the skin, but not other peripheral organs [44].

Expression of anandamide degrading enzyme FAAH decreases with age and in response to sunburn in skin [45], indicating that ECS may undergo similar changes upon skin aging and in response to UV damage. A decrease in FAAH with age may also mean that there is less anandamide produced with age.

There is even less information about ECS activity in other tissues. One report shows a 2-AG decrease in lungs and increase in blood, while AEA increases in lung and blood in mice [46].

## 3. ECS and Cancer

### 3.1. Changes in ECS in Cancer

The metabolic abnormality of lipids has been linked to cancer due to their crucial regulatory roles in signaling pathways involved in initiation and progression of malignancies. The ECS is a biological system comprised of lipid-derived endocannabinoids, cannabinoid receptors, and the enzymes responsible for endocannabinoid metabolism. The ECS is dysregulated in numerous diseases, including cancer. In this section, we will discuss changes in the ECS in human malignancies and the impact on cancer progression and patients’ prognosis. We will also discuss signaling pathways that mediate antitumorigenic or protumorigenic effects of the ECS activation. We will summarize the data for major cancers in Table 2.

#### 3.1.1. Changes in Expression Pattern of Cannabinoid Receptors in Cancer

As mentioned above, cannabinoids exert their biological actions primarily through the activation of various receptors, and many of them are likely to be altered in cancer. In this review, we will mainly focus on three well-characterized G-protein-coupled receptors: cannabinoid receptor 1 (CNR1, also known as CB1R and CB1), cannabinoid receptor 2 (CNR2, also known as CB2R and CB2), and orphan G-protein-coupled receptor 55 (GPR55) [80,81].

##### CB1R and CB2R

To date, many studies using immunohistochemical staining, Western blotting, qRT-PCR, or a combined method have demonstrated overexpression or expression of CB1R and/or CB2R in human cancers, including glioma [75,76,77,78], lymphoma [82,83], leukemia [84,85], breast [64,65,66,67], lung [60,61], ovarian [86,87], pancreatic [88], prostate [89,90,91], skin [52,92,93] and thyroid cancers [94], endometrial [95], esophageal [96], head and neck [97], hepatocellular [98,99,100], renal [101,102], and mobile tongue carcinomas [39,103]. In addition, CB1 and CB2 receptors were highly expressed in non-Hodgkin’s lymphoma, and CB1 in mantle cell lymphoma compared to reactive lymph nodes [83,104]. Correlation also exists between the CB2 receptors expression and estrogen and progesterone receptor, as well as ERBB2/HER-2 levels in breast cancer [105]. Another study showed higher expression of CB1 receptors in androgen-sensitive and androgen-independent prostate cancer cell lines compared to normal prostate epithelial cells [90].

A large body of evidence has indicated that the overexpression of CB1R or CB2R is correlated with reduced survival, increased risk of metastasis and recurrence, and poor prognosis and clinical outcomes [64,65,75,77,84,87,91,94,96,97,102]. Analysis of astrocytoma showed that expression of CB2 receptors correlates with tumor malignancy [75]. The ratio of CB2 to CB1 expression in gliomas correlates with the tumor grade [75]. Additionally, the over-expression of CB1 and TRPV1 correlated with increased grades of prostate tumors [89]. In pancreatic tumors, the over-expression of CB1 was associated with shorter survival rates [106]. Interestingly, several studies have also shown an association between the elevated expression of CB1R and/or CB2R and a better prognosis in hepatocellular carcinoma [99], ERα− and ERα+ breast cancer [66], and non-small-cell lung cancer [60]. Additionally, in hepatocellular carcinoma, the higher expression of CB1 and CB2 correlated with improved prognosis [99]. Importantly, numerous studies have also indicated a downregulation of CB1R and/or CB2R in a few human cancer types, including glioma [79], colorectal cancer [107,108], endometrial [95], hepatocellular [100], and renal cell carcinomas [109]. These data suggest cannabinoid receptors are potential prognostic indicators for cancer patients and should caution us that such indicators are very cancer specific.

In contrast with the extensive studies that have been conducted to determine the expression of cannabinoid receptors in cancers and the correlation with disease clinicopathological parameters and patients’ prognosis, the analysis of mechanisms underlying dysregulation of these receptors has drawn much less attention. Hypermethylation of CpG islands has been revealed in transcription factor binding sites in the *CNR1* promoter of colorectal cancer cell lines and tissue samples examined [108]; it was further found that inhibition of DNA methyltransferase profoundly elevates the *CNR1* transcription, suggesting a crucial contributing role of *CNR1* promoter hypermethylation in downregulating CB1R expression in colorectal cancer. Another study has validated the involvement of DNA methylation in suppressing *CNR1* transcription in the same cancer type [57]. In addition to DNA methylation, miRNAs (miRs)—well-characterized small non-coding RNA molecules (20–22 nt)—play pivotal roles in many biological and pathological processes, including cancer. Therefore, it is not surprising if these small RNA molecules also contribute to the dysregulation of cannabinoid receptors in cancer. The LoVo colorectal cancer cells overexpress miR-1273g-3p, which directly targets *CNR1* [110], leading to activation of the Erb-B2 receptor tyrosine kinase 4 (ERBB4)/phosphoinositide-3-kinase regulatory subunit 3 (PIK3R3)/mechanistic target of rapamycin (mTOR)/S6 kinase 2 (S6K2) pathway, eventually promoting the malignant behavior of LoVo cells. Another study has shown that miR-23b-3p and miR-130a-5p, which are downregulated in gastric cancer cells, directly silence CB1R, attenuating cell growth, migration, and invasion of the cancer cells [63].

##### GPR55

Growing evidence has demonstrated that GPR55 is overexpressed in numerous malignancies, including breast [68,69] and colorectal cancers [57,59], endometrial [111] and squamous cell carcinomas [112]. The elevated expression of GPR55 is significantly correlated with metastasis, reduced disease-free survival, and poor prognosis in breast cancer [68,69].

Mechanically, although *GPR55* does not have a CpG island, its DNA methylation is globally reduced in colorectal cancer [57], which may contribute to the elevated transcription of *GPR55* gene. GPR55 mRNA is a direct target of miR-675-5p [113]. Downregulation of miR-675-5p has been shown in non-small-cell lung cancer (NSCLC), which is correlated with TNM stage and lymph node metastasis of this disease [113], and may play a role in the upregulation of GPR55 expression in NSCLC. Overexpression of miR-675-5p inhibits tumor growth, proliferation, migration, and invasion of NSCLC cells in vivo and in vitro, by targeting *GPR55* [113].

#### 3.1.2. Changes in Cannabinoid Receptor Endogenous Ligands in Cancer

Anandamide (AEA) and 2-arachidonoylglycerol (2-AG) are the two most bioactive endocannabinoids [114], which trigger activation of CB receptors and regulate downstream signaling pathways in a receptor-dependent or -independent manner. Growing evidence has indicated differential regulation of endocannabinoids AEA and 2-AG in numerous cancers.

The AEA and 2-AG were increased in many cancer cell lines and tissues, including glioblastoma, meningioma, pituitary adenoma, endometrial sarcoma, prostate, and colon carcinoma [70,71,72,73,74]. Additionally, the concentrations of 2-AG and AEA were higher in CRC cells than in healthy neighboring tissues [71,73,115,116]. The levels of plasma and/or tissue 2-AG are elevated in colorectal cancer [55], craniopharyngioma [117], diffuse large B-cell lymphoma [118], glioma [78], and hepatocellular carcinoma [100], whereas the levels of AEA are reduced in hepatocellular carcinoma [100] and glioma [78]. In consistence with downregulation of AEA in glioma and hepatocellular carcinoma [78,100], the functional studies support a tumor suppressive role of AEA in head and neck and laryngeal squamous cell carcinomas [119,120]. Although 2-AG was elevated in various human malignancies [55,78,100,117,118], the functional evidence does not support an oncogenic role of 2-AG in tumor progression. Indeed, 2-AG exhibits an anticancer property in several model systems. 2-AG suppresses pancreatic cancer cell proliferation and tumor growth in vitro and in vivo [121], which could be blocked by CB1R antagonist, but not CB2R antagonist, indicating that 2-AG-induced antiproliferative effect is CB1R dependent. 2-AG also inhibits proliferation of diffuse large B-cell lymphoma and laryngeal squamous cell carcinoma cells [118,120], although it has been shown to promote proliferation of a few diffuse large B-cell lymphoma cell lines [118].

Lysophosphatidylinositol (LPI) is an endogenous agonist of GPR55 receptor. The levels of LPI have been shown to be elevated in two cancer types: breast [63] and colorectal cancers. Functional studies have indicated that LPI profoundly promotes migration/invasion of breast and colorectal cancer cells, which could be blocked by GPR5 antagonist or GPR55 knockdown [56,63,122]. Furthermore, LPI produced by ovarian cancer cells promotes angiogenesis [123], which is prevented by GPR55 inhibition or GPR55 knockdown. Interestingly, N-docosahexaenoyl dopamine triggers apoptosis of cancer cells via activation of GPR55 [124], and LPI enhances the cytotoxic effect. These data suggest that the antiproliferative effect of endocannabinoids and the prometastatic effect of LPI are mainly mediated by cannabinoid receptors.

#### 3.1.3. Changes in the Expression Pattern of Endocannabinoid Hydrolytic Enzymes in Cancer

AEA and 2-AG are synthesized from the substances N-arachidonoyl phosphatidylethanolamide (NAPE-PLD) and diacylglycerol (DAG) by phospholipase D and DAG lipase, respectively, and degraded by FAAH and MAGL, respectively. A large body of evidence has demonstrated that aberrant expression of enzymes responsible for endocannabinoid metabolism results in dysregulation of endocannabinoid metabolism that may drive progression of cancer [125,126,127]. Here, we will mainly discuss changes in FAAH and MAGL expression in cancer and their impact on cancer biology and prognosis.

FAAH is overexpressed in prostate cancer cell lines and tumor tissues; tumor FAAH immunoreactivity (FAAH-IR) is positively correlated with disease severity for cases with mid-range CB1 expression [128]. The high tumor FAAH-IR may serve as an indicator of poor prognosis [128,129]. Functional studies have shown that downregulation of FAAH in prostate cancer cells attenuates 2-AG hydrolysis and cell invasion [130], and the phenotypic effects can be reversed by FAAH overexpression. Furthermore, pharmacological inhibition of FAAH suppresses invasion and metastasis of lung cancer and colon adenocarcinoma cells in vivo and/or in vitro [62,131]. Moreover, pharmacological inactivation of CB2R attenuates FAAH-siRNA-induced TIMP-1 expression, suggesting a role of CB2R in mediating the anti-invasive and anti-metastatic signaling triggered by FAAH inhibition [62]. Data from The Human Protein Atlas (https://www.proteinatlas.org/ (accessed on 15 June 2022)) indicate that FAAH expression increases in most of the studied cancers, with the most drastic increase in prostate cancer [45]. These data suggest that FAAH may act as an oncogene. Interestingly, several lines of evidence have also indicated downregulation of FAAH in tumors, including endometrial carcinoma, glioma, and uterine leiomyoma [78,132,133].

Numerous studies have shown an overexpression of MAGL in various human malignancies, including prostate adenocarcinomas [130], malignant melanoma [53], osteosarcoma, hepatocellular carcinoma [134,135], cervical [136], colorectal [54], and endometrial cancers [137]. The overexpression of MAGL is correlated with larger tumor size, vascular invasion, poor differentiation, and clinicopathological stage of several cancers [53,134,135,137]. Therefore, the elevated MAGL may act as an indicator of poor prognosis in cancer patients. Functional studies have shown that inhibition or knockdown of MAGL significantly suppresses proliferation, migration, invasion, and xenograft tumor growth in vitro and/or in vivo [54,134,136,137,138]; knockdown of MAGL also induces apoptosis and cell cycle arrest via downregulation of cyclin D1 and Bcl-2 [54,137], or upregulation of Bax and cleaved caspase-3 [136]. Conversely, overexpression of MAGL promotes tumor growth, migration, invasion, and metastasis in nitro and/or in vivo [134,139,140,141], via a NF-κB-mediated EMT process. These data suggest an oncogenic role of MAGL in malignant progression.

### 3.2. Changes of ECS Signaling Pathways in Cancer—Potential Molecular Targets

Alterations in all components of endocannabinoid system contribute to cancer progression through multiple pathways of cellular signaling. Below, we will discuss ECS changes through receptor-dependent and -independent signaling.

#### 3.2.1. Receptor-Dependent Signaling and Changes in Response to Cannabinoids

GPCRs are the biggest family of receptors targeted by approved drugs [142]. However, they are rarely targeted in cancer therapy, with the exception of endocrine tumors (pituitary, adrenal, testes, ovarian) and hormone-dependent tumors of breast and prostate. Although mutations in GPCRs are not often the “driver mutations”, they take a large part in regulation of cell signaling, which regulates cellular functions such as metabolism, growth, and proliferation [143]. Considering that CB receptors belong to the GPCR family of receptors with a wide variety of cell downstream signaling, they are likely the best therapeutic targets in cancer patients.

Pharmacological activation or upregulation of CB1R and/or CB2R inhibits tumor growth and metastasis in vivo, attenuated proliferation, migration/invasion, and angiogenesis, and induced apoptosis in vitro through inhibition of extracellular signal-regulated kinase (ERK), protein kinase B (PKB)/AKT, cAMP/protein kinase A (PKA), c-Jun N-terminal kinase (JNK), MAPK p38, nuclear factor kappa B (NF-κB), and/or vascular endothelial growth factor (VEGF) signaling pathways, in various human cancers, including glioma [144,145], leukemia [146,147,148], multiple myeloma [149], neuroblastoma [150], breast [151,152,153,154], cervical [105], colorectal [110,155,156,157], endometrial [158], hepatocellular [159], intestinal [108], non-small-cell lung [160,161], prostate [162,163], and thyroid cancers [164]. The antitumorigenic and/or proapoptotic effects can be blocked by inhibition and/or knockdown of CB1R and/or CB2R [144,145,146,149,152,153,155,160,162,164]. This is very interesting, since CB1 and CB2 are overexpressed is many cancers as well. The level of endocannabinoids is also often upregulated in cancers, as well as the level of FAAH and MAGL. So, it is counterintuitive that one can inhibit growth of cancer cells by overexpressing or/and activating the CB1/CB2 receptors or adding endocannabinoids.

It has been reported that the receptor-triggered antitumorigenic and/or proapoptotic effect can be mediated via activation of signaling pathways. Activation of CB1R and CB2R profoundly inhibits malignant glioma growth in rat and mouse models through activation of ERK pathway [165]. Pharmacological activation of CB1R and CB2R induces apoptosis of hepatocellular carcinoma cells through upregulation of proapoptotic factors Bax and Bcl-x(s) and downregulation of antiapoptotic factors Bcl-2 and survivin, through activation of JNK/p38 MAPK pathway [166]. Interestingly, stimulation of cannabinoid receptors may have a dual effect on signaling pathways. In addition to attenuating the Akt pathway, activation of CB1R and/or CB2R also causes activation of JNK [108,162] and ERK1/2 [163] pathways in prostate and non-small cell lung cancer cells.

Evidence has also demonstrated both protumorigenic and anti-apoptotic roles of the ECS. Pharmacological inhibition of CB1R attenuates proliferation and tumor growth, inducing apoptosis and cell cycle arrest in breast cancer through upregulation of p27^KIP1^ and downregulation of cyclin D and E [167]. The elevated expression of cannabinoid receptors induced by chronic intermittent hypoxia promotes tumor growth, migration, angiogenesis, and lung metastasis of breast cancer cells both in vitro and in vivo [168], through activation of insulin-like growth factor-1 receptor (IGF-1R)/AKT/glycogen synthase kinase-3β (GSK-3β) pathway, which can be blocked by knockdown of cannabinoid receptors. Furthermore, pharmacological activation of CB2R enhances proliferation and tumor growth of colon cancer cells in vitro and in vivo via activation of AKT/PKB pathway [169]. Knockdown of CNR1 diminishes proliferation and migration of progesterone-resistant endometrial cancer cells, and resensitizes these cells to progesterone, through inhibition of ERK and NF-κB pathways [158]. Moreover, activation of CB1R prevents astrocytoma cells from ceramide-induced apoptosis through activation of ERK pathway [170]. Inhibition or knockdown of cannabinoid receptors attenuates proliferation, migration, and induces apoptosis of HPV+ head and neck squamous cell carcinoma cells [171]. Conversely, pharmacological activation of CB1R and CB2R enhances cancer cell growth and migration, and attenuates apoptosis both in vitro and in vivo, via activation of p38 MAPK pathway [171].

GPR55 receptor also plays a pivotal role in tumor progression. Stimulation of GPR55 promotes cancer cell proliferation and tumor growth in vitro and in vivo via activation of ERK pathway [172]. Overexpression of miR-675-5p inhibits tumor growth, attenuates proliferation, migration, and invasion, and induces G1 cell cycle arrest of non-small-cell lung cancer cells both in vitro and in vivo via direct targeting of GPR55, through inhibition of ERK pathway [113]. Furthermore, the LPI-mediated activation of GPR55 promotes proliferation of ovarian and prostate cancer cells [80] and ovarian-cancer-cell-induced angiogenesis [123], via activation of Akt, ERK1/2, and p38 MAPK pathways. Moreover, pharmacological inhibition of GPR55 enhances doxorubicin cytotoxicity in cancer cells through inactivation of MEK/ERK and PI3K/AKT pathways [173]. These data indicate that GPR55 may function as an oncogene. Interestingly, pharmacological activation of GPR55 in cholangiocarcinoma cells, however, diminishes cancer cell proliferation through activation of JNK pathway [174], which can be blocked by GPR55 knockdown, suggesting that GPR55 may also act as a tumor suppressor.

#### 3.2.2. Receptor-Independent Signaling and Changes in Response to Cannabinoids

The ECS is also able to exert its biological and pathological effects in a receptor-independent manner. Pharmacological activation of cannabinoid receptors enhances the radiation-mediated anti-proliferative effect on breast cancer cells via a receptor-independent sphingosine-1-phosphate (S1P)/ceramide pathway [175]. CBD triggers apoptosis of breast cancer cells by attenuating mitochondrial membrane potential and promoting the release of cytochrome c, eventually leading to activation of a receptor-independent intrinsic apoptotic pathway [176]. Interestingly, cannabinoids triggers apoptosis only in astrocytoma cells expressing low levels of cannabinoid receptors through activation of ERK1/2 pathway, not in the cells overexpressing the receptors [177]. Furthermore, AEA induces apoptosis of pheochromocytoma cells through activation of p38 MAPK/JNK pathway, which cannot be rescued by CB1R antagonist [178]. Moreover, the synthetic cannabinoid CP55940 induces apoptosis and cell cycle arrest of T-cell acute lymphoblastic leukemia cells by upregulation and activation of proapoptotic molecules via activation of c-Jun/JNK pathway [179]. However, the CP55940-induced apoptosis in leukemia cells cannot be rescued by cannabinoid receptor agonists, suggesting involvement of cannabinoid receptor-independent mechanism. These data indicate that cannabinoids can trigger programmed death of cancer cells via receptor-independent signaling.

#### 3.2.3. Signaling When the Receptor Status Is Unknown

Numerous studies have also shown the antitumorigenic and/or proapoptotic effects of cannabinoids on human malignant cells, while not reporting the involvement of cannabinoid receptors in these processes. CBD profoundly attenuates proliferation and invasion of breast cancer cells in vitro through inactivation of EGFR, AKT, ERK, and NF-κB pathways [180]. Consistently, in a mouse model, CBD also suppresses tumor growth and metastasis of breast cancer cells via attenuation of macrophage recruitment to xenograft tumor sites [180]. Furthermore, CBD enhances radiation-induced glioblastoma cell death through inhibition of ERK1/2 and AKT pathways, and activation of JNK1/2 and p38 MAPK pathways [181].

Interestingly, endocannabinoids act differentially on tumor growth. AEA inhibits tumor growth of cholangiocarcinoma cells via upregulation and activation of Notch1 [182], whereas 2-AG promotes tumor growth through upregulation and activation of Notch2. The antitumorigenic effect of AEA on cholangiocarcinoma may also require activation of Wnt-JNK pathway [183], since that can be abolished by Wnt5a knockdown.

Recently, using two neuroblastoma cell lines as a model system, we have revealed a suppressive role of cannabinol (CBN) on neuroblastoma cell proliferation, invasion, and angiogenesis through miR-34a-mediated targeting via inhibition of AKT pathway [184]. We have found that a novel 31 nt tRNA_i_^Met^ fragment tRiMetF31 generated from miR-34a-guided cleavage [185] can directly target 6-phosphofructo-2-kinase/fructose-2,6-biphosphatase 3 (PFKFB3), a key proangiogenic factor, and highlighted a crucial role of the miR-34a/tRiMetF31/PFKFB3 axis in CBN-mediated suppression in neuroblastoma biology. We have not studied the role of cannabinoid receptors in this process, however.

## 4. Effect of Cannabinoids on Various Hallmarks of Cancer

Various in vitro and in vivo experiments have shown that cannabinoids can target almost every hallmark of cancer (Figure 2) [186]. They inhibit proliferation, reduce inflammation, stimulate apoptosis, and inhibit tumor invasiveness, angiogenesis, and metastasis [187,188,189,190]. One of the most important effects of cannabinoids, besides their antitumor ability, is that they are less likely to affect non-transformed normal cells surrounding tumors, and they may even have protective effects. For instance, cannabinoids may induce cell death in glioma cells while protecting normal astroglial and oligodendroglial cells from apoptosis via CB1 receptors [187]. Studies on animals show the protective effects of cannabinoids against certain types of tumors. For example, a dose-dependent decrease in the incidence of hepatic adenomas and hepatocellular carcinomas in mice that were given THC over 2 years was noted. Additionally, lower incidence rates of benign tumors in mammary glands, uterus, testis, and pancreas were seen in tested rats [191].

### 4.1. Induction of Autophagy and Apoptosis

Autophagy and apoptosis are two essential mechanisms of regulation of uncontrolled growth. Autophagic activity of cannabinoids observed in several major cancers [192,193] is partially dependent on the CB1 or CB2 receptor. Mice deficient in CB1 receptor exhibit altered autophagosomal activity [13], while endocannabinoid palmitoylethanolamide increased the phagocytosis of murine microglial cells [194]. Additionally, the experimental study using delta-9-THC and a synthetic agonist decreased the cell viability of hepatocellular carcinoma xenografts in nude mice via the CB2 receptors. The anti-cancer effect was explained by activating the endoplasmic reticulum stress response, which leads to macro-autophagy and eventually apoptosis [195]. Studies on small-cell lung carcinoma [61] and breast cancer cells [67] supported the idea that CB1 and CB2 receptors may be potential targets to achieve apoptosis. The preclinical models of breast cancer showed evidence that CBD may induce apoptosis in estrogen-dependent and estrogen-independent breast cancer cells with little or no effect on normal mammary cells. Surprisingly, this was CB1-, CB2-, and vanilloid receptor-independent [176].

The well-established antineoplastic mechanisms of cannabinoids are alterations in ceramide de novo synthesis. In cancer cells, increased ceramide levels, a neutral lipid backbone of complex sphingolipids, can occur under chemotherapy, radiation, and stimulation of CB receptors [58,196]. As a result, ceramide activates endoplasmic reticulum stress response and causes inhibition of global translation of proteins. At the same time, there is an activation of C/EBP homology protein (CHOP) which can stimulate proapoptotic proteins BAD and BAX [197]. Moreover, cannabinoids can cause downregulation of AKT, which may have a variety of intracellular effects. Low AKT leads to activation of autophagy via the mTOR pathway, cell cycle arrest through p21, and activation of caspase 9 and 3, which eventually ends in apoptotic cell death [58,108,155,198,199,200].

Activation of CB1 and CB2 receptors by synthetic cannabinoid agonists could stimulate apoptosis via ceramide synthesis and TNF-receptor activation [58]. Another group showed that activation of CB1 receptors in different CRC cell lines causes inhibition of major cancer survival pathways such as RAS/MAPK, ERK1, and PI3K/AKT [155]. Additionally, CBD, a partial agonist of CB1/CB2 receptors and antagonist of GPR55, may suppress mTOR/AKT signaling and activate proapoptotic NOXA in CRC cells [201]. Moreover, CBD suppressed the production of inhibitors of apoptosis, such as survivin and c-FLIP in colon cancer cells [197].

### 4.2. Reduction of Inflammation and Inhibition of Proliferation

Inflammation is a large component of carcinogenesis. ECS plays a central in the regulation of function of immune system and control of inflammation. Similarly, many phytocannabinoids exert strong anti-inflammatory effects upon local [202] or systemic [203] application.

Cannabinoids inhibited proliferation by suppressing the AKT/PKB prosurvival pathway causing cell cycle arrest in G1/S phase. This was shown in multiple cancers, including melanoma, breast, gastric, lung, and liver carcinomas [93,151,159,160,204,205]. In a breast cancer model, cannabinoids were able to induce cell cycle arrest via inhibition of cyclin dependent kinase 1 (CDK1), induction of p21 and p27, a decrease in cyclin A and E levels, degradation of CDC25A, and finally, inactivation of CDK2 [206,207].

In the study on head and neck squamous cell carcinoma, cannabinoids were able to stimulate dual specificity phosphatase 1 (DUSP1), which is a negative regulator of MAPK [208]. DUSP1 is one of the central mediators in the resolution of inflammation in cells. Moreover, the levels of cyclin dependent kinase inhibitor, p21, as well as growth arrest and DNA damage-inducible protein α (GADD45A) were activated, resulting in cannabinoid’s antiproliferative effects. In human gastric cancer model, CBD upregulated ATM and p21, which caused a decrease in CDK2 and CCNE, resulting in cell arrest in G0/G1 stage [209]. In a xenograft model of human glioma, CBD was able to reduce the activity of 5-lipoxygenase, an enzyme that catalyzes synthesis of leukotrienes (LTs) and mediators of inflammation; a decrease in 5-lipoxygenase activity caused inhibition of LTB4 and had antiproliferative effect [210].

The eicosanoid system, which contains pro- and anti-inflammatory molecules, plays an important role in cannabinoid-induced tumor cell apoptosis. The addition of R(+)methanandamide to the glioma cells activated de novo ceramide synthesis, which eventually led to COX-2 expression with subsequent production of PGE2 that had proapoptotic effect [211,212]. It was shown that proapoptotic effects of eicosanoids was PPARγ receptor-dependent [213,214,215].

On the other hand, it was shown that the micromolar concentrations of THC, CB1 agonist—arachidonyl-2-chloroethylamine (ACEA), and CB2 agonist HU308 stimulated the proliferation of cancer cells, which can be explained by transactivation of EGFR [169,171,216,217].

The chemoprotective effect of CBD was also shown on colorectal cancer in mice. Adding CBD prevented premalignant and malignant lesions development in the azoxymethane model of colon cancer [218]. The effect was explained by DNA protection against oxidative damage, increased levels of endocannabinoids, and decreased cell proliferation [218]. The antiproliferative action was CB1 dependent [219].

### 4.3. Inhibition of Angiogenesis, Tumor Invasiveness, and Metastasis

There were multiple reports showing the inhibitory effects of cannabinoids on cancer cell migration, invasion, and metastasis [62,74,220]. CBD was shown to inhibit the invasiveness of lung cancer cell lines by inhibiting ICAM-1 [190]. As some experiments indicated, the induction of tissue inhibitor of metalloproteinase-1 (TIMP-1), and ICAM-1 by THC, Met-AEA and CBD had significant anti-invasive effects [105,190,221]. The action of TIMP-1 is achieved via reduction of collagen-degrading enzymes, MMP-2 and MMP-9, that promote cancer cell invasiveness [222].

Another way in which cannabinoids are diminishing tumor aggressiveness is inhibition of epithelial-to-mesenchymal transition. A study that involved 2-methyl-2′-F-anandamide (Met-F-AEA) showed a significant reduction in β-catenin, vimentin, N-cadherin, and fibronectin, which are considered mesenchymal markers in tumor invasion. Moreover, Met-F-AEA decreased the levels of EMT markers such as Snail1, Slug, and Twist [223]. Other studies showed that CBD may reverse an IL-1β-induced EMT in breast cancer cells [224], or TGF-β-induced reorganization of F-actin, which also corresponds to EMT in lung cancer cells [60]. Cannabinoids may inhibit the invasion and metastasis of cancer cells through downregulation of vascular endothelial growth factor (VEGF), matrix metalloproteinase 2, matrix metalloproteinase 9, E-cadherin, cyclooxygenase 2 (COX-2), and hypoxia-inducible factor α [225,226,227].

## 5. Effect of Terpenes and Flavonoids

Some preclinical studies have shown that *Cannabis* extracts may be more effective than cannabinoids alone for cancer treatment. For instance, high-CBD extract showed higher affinity for CB1 and CB2 receptors than CBD alone. As a result, a high-CBD extract was more potent in preventing intestinal polyps’ formation in animal models [219,228].

The cannabis plant is rich in terpenes and flavonoids, biologically active substances which can also be used in cancer treatment [229,230]. There are more than 20,000 terpenes in nature, with around 200 found in *Cannabis* plants [231]. The monoterpene myrcene, sesquiterpenes β-caryophyllene, and α-humulene are often present in *Cannabis* chemovars. However, the spectrum of terpenes can vary from plant to plant. We will describe only some of the common terpenes that have anti-neoplastic effects.

Myrcene is present in hop, bay, verbena, lemongrass, citrus, and even carrot. Surprisingly, in some animal studies, myrcene showed to be carcinogenic, causing kidney cancer in rats and liver cancer in mice [191]. Another study showed that myrcene protected human B lymphocytes from DNA damage caused by hydroperoxides [232]. However, it also had cytotoxic effects on breast, colon, cervical, lung cancer cell lines, and leukemia cells [231,233]. There is not much knowledge about the mechanisms of action of myrcene, and more studies should be undertaken considering its controversial effects on cancer cells.

β-caryophyllene is a sesquiterpenoid commonly present in black pepper, oregano, basil, and rosemary. This terpene can induce apoptosis and cause cell cycle arrest in lung and ovarian cancer cell lines [234,235]. It can also influence the production of free radicals and can have antiapoptotic and antiproliferative effects via activation of the JAK1/STAT3 pathway in osteosarcoma cells [236]. Importantly, β-caryophyllene may sensitize different cancer cell lines to conventional chemotherapy drug doxorubicin [237,238,239,240]. Additionally, it attenuated doxorubicin-induced chronic cardiotoxicity in rats via activation of CB2 receptors [241]. Moreover, the combination of β-caryophyllene with 5-fluorouracil (5-FU) or oxaliplatin on colorectal cancer cells sensitized those cells to chemotherapeutics [242]; similarly, combination of β-caryophyllene with sorafenib potentiated the effect on liver cancer cells [243]. Thus, combining different cannabinoids with β-caryophyllene may become advantageous in cancer therapy, which needs further investigation.

The monocyclic sesquiterpene, humulene, has cytotoxic activity on multiple cancer cell lines via increasing production of reactive oxygen species [244,245] and inhibition of AKT in hepatocellular carcinoma cells with activation of apoptosis [246]. In in vitro models, humulene enhanced 5-fluorouracil, oxaliplatin, and doxorubicin cytotoxic effects [239,242].

Another terpene, limonene, is a cyclic monoterpene mainly present in citrus plants and is also present in cannabis. In the bladder cancer cell line, limonene caused G2/M cell cycle arrest, decreased migration, and metastasis, and increased Bax and caspase 3, thus inducing apoptosis [247]. It inhibited PI3K/AKT, induced autophagy and enhanced sensitivity to docetaxel in in vitro cancer cell models [248,249,250,251]. In in vivo models, limonene decreased tumor growth, induced apoptosis, and reduced c-Jun and c-myc expression [251,252,253,254,255,256,257,258,259]. There was one small clinical trial in which breast cancer patients received limonene for a short period of time; limonene decreased cell cycle regulatory protein expression, including cyclin D1 in breast cancer patients [260].

Pinene is present in pine resins, rosemary, basil, and parsley. As multiple preclinical data show, pinene was able to reduce the cell viability, stimulate apoptosis, and induce cell cycle arrest in numerous cancer cell lines [261,262,263,264,265,266,267]. Moreover, it can act synergistically with paclitaxel in tested lung cancer [265]. In in vivo animal models, pinene showed reduced growth and number of tumors [268].

These data could also support the advantageous action of cannabis extracts rich in terpenes versus purified cannabinoids in fighting against different malignancies. Different modulatory (often referred to as “entourage”) effects of cannabinoids and other substances in the cannabis plant were extensively reviewed in the past [269], although exact mechanisms are still unclear.

## 6. Preclinical and Clinical Use of Cannabinoids

### 6.1. Cannabis and Cannabinoids for Primary Care—Tumor Shrinkage

#### 6.1.1. Data on Humans Are Limited

Cannabis has been used for medicinal purposes for thousands of years, until the 1940s, when the authorities prohibited it. In the USA, cannabis is classified as a Schedule I agent with risk for abuse and no approved medical use [270]. Despite the solid preclinical evidence regarding the antitumor properties of cannabinoids, there were not many human trials that supported this effect of cannabinoids (see Table 3 and Table 4). This could partially be because of the multiple legislation problems with cannabis. Thus, it is still kept on “the shelf” as a backup medication, mainly for palliative care in cancer patients. The number of clinical studies related to the role of cannabis and cannabinoids in cancer is critically low (see Appendix A). Today, there are few human trials regarding the palliative effects of cannabinoids in cancer patients, and even fewer regarding their anti-cancer effects (based on the clinicaltrials.gov database, 22 June 2022).

There were few human studies regarding delta-9-tetrahydrocannabinol (THC) in brain tumors. One of the first human trials involving cannabis as a cancer treatment was a small study on recurrent glioblastoma with intra-tumoral injections of delta-9-THC. Unfortunately, this study showed no beneficial effect [288]. However, a case report in two children with pilocytic astrocytoma after subtotal resection showed spontaneous regression of the tumor 3 years after the surgery. The patients did not receive any conventional adjuvant therapy, but they had inhalations of cannabinoids [312]. Another trial showed that the combination of temozolomide and Sativex increased 1-year survival rates in glioblastoma multiforme patients (NCT01812603, NCT01812616). In a pilot I study in nine glioblastoma patients that failed conventional therapies and had signs of tumor progression, patients received intratumoral injections of THC. Results showed the reduction of Ki67 immunostaining and antiproliferative effect of THC on tumor cells [288]. Another study in 2 glioblastoma patients receiving intratumoral injections of THC showed THC effectiveness in reducing VEGF and VEGFR-2 activation [145]. In the case study of a 14-year-old patient with an aggressive form of acute lymphoblastic leukemia, remission was achieved following the consumption of hemp oil, after bone marrow transplant, aggressive chemotherapy, and radiotherapy were revoked. A double-blind study tested oromucosal spray containing nabiximols in conjunction with temozolomide. Overall, the 1-year survival rates were higher in patients using nabiximols (83%) compared to the placebo group (44%) [289].

One interesting study included CBD as an immunosuppressive and anti-inflammatory agent in the adjunct treatment of acute graft-versus-host disease (GVHD) after allogeneic hematopoietic stem cell transplantation in 48 patients with acute leukemia and myelodysplastic syndrome. The results of the study showed that the combination of CBD with conventional graft-versus-host disease prophylactic treatment was safe and showed a lower incidence of GVHD [290].

#### 6.1.2. Combination of Cannabinoids with Other Drugs—There Is Potential Benefit, but Caution Is to Be Exercised

The combination of cannabinoids with conventional anti-cancer therapy is also under investigation. For instance, the combination of cannabis with gemcitabine reduced the cell viability of pancreatic cells in vitro [192].

Moreover, adding THC to temozolomide treatment increased the sensitivity of chemotherapy-resistant glioma cells to the treatment in mice models [313]. Another study showed that the combination of THC with CBD enhanced radiation’s effects on the murine glioma model [314]. CBD may also overcome the oxaliplatin resistance of cancer cells via inhibition of superoxide dismutase 2 and activation of autophagic response [315]. The combination of CBD with a conventional chemotherapy agent, carmustine, caused inhibition of proliferation in glioblastoma multiforme cell line and overcame the carmustine resistance via activation of TRPV2 [316]. Additionally, the combination of CBD with THC showed higher antiproliferative action on glioblastoma multiforme cell lines. Furthermore, CBD stimulated TRPV2 and increased uptake of cytotoxic drugs by glioma cancer cells without affecting normal astrocytic cells [144]. The combination of THC with CBD also enhanced the action of temozolomide in mouse models [200,313].

Before cannabinoids can be prescribed as an actual treatment in cancer patients, their pharmacokinetics should be considered. In vitro studies showed that CBD can inhibit cytochrome P450, which is responsible for the metabolism of many medications, including conventional chemotherapeutics. As a result, a high concentration of CBD may increase the toxicity and decrease the potency of standard anti-cancer therapy [317,318]. Thus, the interaction of cannabinoids with cytochrome P450 raised a valuable concern about combining it with conventional chemotherapeutics. A clinical study involving 24 patients receiving irinotecan or docetaxel that were using cannabis showed that the addition of cannabis tea did not significantly affect clearance and medication exposure [319]. However, there is an obvious need for more data regarding cannabinoid pharmacokinetics and interaction with other medications, as many cancer patients are using cannabis for different purposes.

A significant concern for using cannabis with anti-cancer treatment was raised in patients undergoing immunotherapy. A retrospective observational study evaluated the influence of cannabis during nivolumab therapy in 140 patients with advanced melanoma, non-small-cell lung cancer, and renal cell carcinoma [310]. In this study, 89 patients received nivolumab and 51 received nivolumab and cannabis. As a result, cannabis reduced the response rate to immunotherapy. It was shown that the response rate to nivolumab alone was 37.5%, and for nivolumab, with cannabis, it was only 15.9%. However, there was no difference in overall survival [310]. Another prospective observational study from the same group followed 102 patients with metastatic cancers that started immunotherapy. In this study, 68 patients received immunotherapy, and 34 were on immunotherapy while using cannabis. Participants using cannabis had 39% of clinical benefit, whereas patients receiving immunotherapy had 59%. Moreover, in a cannabis arm, the tumor progression took 3.4 months compared to nonusers, for whom it took 13.1 months. The overall survival in cannabis users was only 6.4 months, and for patients on immunotherapy, it was 28.5 months. These results may be related to the immunosuppressive effects of cannabinoids, and cannabis use should be carefully considered in patients on immune checkpoint inhibitors [311].

### 6.2. Cannabis for Palliative Care

Cancer patients are accessing cannabis to alleviate various symptoms and improve their quality of life. Cannabis medication may reduce the devastating symptoms experienced by cancer patients, such as pain, emesis, anxiety, loss of appetite, and poor sleep quality [320]. A cross-sectional survey of 936 cancer patients revealed that 24% considered themselves active cannabis users. The reasons for cannabis ingestion were to alleviate the physical symptoms such as pain, nausea, and loss of appetite (75%); neuropsychiatric symptoms (63%), recreational use (35%), and cancer treatment (26%) [321]. Thus, the addition of cannabinoids to cancer care seems inevitable regardless of the legislation procedures. However, what if the usage of cannabis as palliative care is affecting the conventional anti-cancer treatment? We already discussed that cannabinoids have anti-cancer properties. However, they can affect drug metabolism and have a negative impact on immunotherapy. Thus, there is a huge need for clinical trials regarding the specific anti-cancer therapy and cannabinoid use to uncover their antineoplastic benefits, as well as to ensure the safe conditions for their ingestion by cancer patients.

#### 6.2.1. Cannabis for Pain

Pain is one of the most devastating symptoms in patients with advanced cancer. It is estimated that eight out of ten patients with advanced cancer experience moderate to severe pain, and around 55% of cancer patients have chronic cancer-related pain. The mechanism of cancer-related pain can be inflammation, invasion of organs, or nerve injury. Opioids are the essential treatment for cancer-related pain; however, they cause addiction, and the overdose can be lethal [322]. Thus, adjusting or lowering the dose of opioids and maintaining an analgesic effect is crucial for cancer patients.

Both cannabinoid and opioid receptors have similar neural transduction systems and are expressed in the parts of the brain responsible for nociception, such as periaqueductal gray, raphe nuclei, and central–medial thalamic nuclei [323]. Moreover, CB1 and μ-opioid receptors colocalize in peripheral pain afferent neurons [324]. The cannabinoid signaling in pain is related to the distribution of CB1 receptors in the spinal dorsal horn. CB1 receptors in presynaptic neurons are colocalized with transient receptor potential cation channel 1. Activation of the CB1 receptor decreases calcium influx, resulting in the lower release of neurotransmitters [325]. The CB1 receptors present in postsynaptic neurons cause an increase in potassium influx that results in hyperpolarization and reduction of neuron excitability [326]. It was also observed that CB2 receptors could indirectly stimulate opioid receptors in afferent pathways, thus enhancing the analgesic effects of opiates [327]. There is also evidence that anandamide, 2-AG, and exogenous cannabinoids can interact with opioid receptors [328]. All these data support the idea that cannabinoids may exert an antinociceptive effect alone and in combination with opiates.

A systematic review and meta-analysis published in 2015 included 79 trials with 6462 participants who assessed different cannabinoid indications, including chemotherapy-induced nausea and vomiting (CINV), appetite stimulation in HIV/AIDS, chronic pain, spasticity due to multiple sclerosis and paraplegia, depression, anxiety, sleep disorder, psychosis, glaucoma, or Tourette’s syndrome, showed the cannabinoid’s effectiveness in pain management [329]. In a systematic review that summarized 28 studies involving Cannabis, dronabinol, nabilone, and nabiximols, 12 included patients with neuropathic pain, 3 with cancer-associated pain, and 1 with chemotherapy-induced pain. The mean number of participants who indicated a reduction in pain of at least 30% was higher with cannabinoids than with placebo. The research showed a significant improvement in cancer-associated pain, with a 1.41 overall odds ratio in two trials [329]. Another review discussed eighteen randomized controlled trials involving 766 patients with chronic non-cancer-related pain and showed that fifteen reported a significant analgesic effect of cannabinoids [330].

Multiple clinical studies showed analgesic properties of cannabinoids in HIV neuropathy, neuropathic pain, spinal cord injury, and diabetic neuropathy [331,332,333,334]. In a meta-analysis of 19 preclinical studies that involved the administration of THC (14 studies), CB1 agonists (3 studies), and CB2 agonists (1 study), 90% demonstrated a statistically significant synergistic effect with opiates. Additionally, the median effective dose of morphine was 3.6, and codeine was 9.5 times lower in combination with delta-9-THC compared to opiates alone. Overall, cannabinoids, when co-administered with opioids, have the opioid-sparing effect, which means that opioid dosage may be reduced without losing its analgesic efficacy [335]. Another randomized controlled trial showed that adding dronabinol to patients on opioids reduced pain and increased patient satisfaction [336]. Moreover, nabiximols reduced pain and improved sleep in cancer patients with poorly controlled pain [337].

One multicenter, double-blind, randomized, placebo-controlled study tested the efficacy of THC/CBD combination and THC extract in patients with intractable cancer-related pain. The study revealed that patients who received THC/CBD extract had a more than 30% reduction in pain according to the Numerical Rating Scale for patients with advanced cancer. In contrast, THC extract showed no significant difference in pain management compared to placebo. The results imply that THC/CBD extract could be an effective adjuvant therapy for cancer-related pain in patients on opioids with inadequate analgetic effect [301].

One of the most disturbing symptoms in cancer patients is neuropathic pain [338]. However, the data involving patients with chemotherapy-induced peripheral neuropathy are very scarce. Preclinical research that modeled vincristine-induced peripheral neuropathy in rats showed that stimulation of CB1 and CB2 receptors prevented the development of neuropathy [339]. Another study involving cisplatin-induced neuropathy in mice presented that the addition of anandamide with the inhibitor of fatty acid amide hydrolase attenuated chemotherapy-induced peripheral neuropathy [340]. Moreover, pretreatment with CBD prevented paclitaxel-induced neuropathy in mice [333]. The only study that involved humans in chemotherapy-induced peripheral neuropathy was a crossover placebo-controlled trial with nabiximols [341]. However, that study reported no significant difference in pain scores between nabiximols and placebo [341].

Although, there were many trials regarding cannabinoids in pain management, their limitations are a small sample size, difficulties with dose adjustment, withdrawal due to adverse effects, and a short duration, which makes it difficult to objectively establish the analgesic durability of cannabis alone, and in combination with opioids. Another issue could be biphasic effect of cannabinoids. It was shown that low doses of THC reduced pain, whereas higher doses exacerbated nociception [320]. Thus, it is critical to gradually titrate the doses of cannabinoids from lower ones to higher and reach a therapeutic window for analgesia whilst avoiding the opposite effect. The presented data shows the necessity in more clinical studies that would be able to optimize cannabinoid ratios and adjust the doses to achieve the maximum analgesic effect.

#### 6.2.2. Reducing Nausea and Vomiting

Nausea and vomiting are prevalent symptoms in cancer patients. Gastrointestinal obstruction, increased calcium levels, metastasis, intoxication, and even the medications prescribed to cancer patients can induce emesis [320]. Despite the wide availability of antiemetics, many cancer patients suffer from chemotherapy-induced nausea and vomiting (CINV), one of the major side effects of conventional anti-cancer therapeutics [342]. Studies suggest that cannabinoids can inhibit nausea physiologically via CB1 and CB2 receptors in the brainstem dorsal vagal complex, which regulates emesis [343]. The preclinical study showed that emesis is controlled by the endocannabinoid system, which is mediated by 5-hydroxytryptamine 3 (5-HT_3_) receptors. The 5-HT_3_ and CB1 receptors are both present on GABA-ergic neurons and have opposite effects on the release of neuromediators [344]. Cannabinoid agonists, including THC, can bind to 5-HT_3_ receptors and antagonize their signaling [328].

Both dronabinol and nabilone were FDA approved in 1985 for CINV in patients refractory to standard antiemetics [345]. However, after the introduction of highly effective 5-HT_3_ receptor antagonists in 1991, which are now standard therapy for acute and delayed CINV, cannabinoids became more of a last resort for CINV treatment [345].

There were multiple clinical trials regarding the antiemetic effects of cannabinoids. A systematic review looking at 30 randomized comparisons of nabilone, dronabinol, or levonantradol in 1366 patients found that cannabinoids are more effective antiemetics than standard prochlorperazine, metoclopramide, chlorpromazine, thietylperazine, haloperidol, domperidone, or alizapride. Across all trials, cannabinoids were more effective than their comparators and placebo when it came to completely controlling nausea and vomiting [346]. However, the side effects of cannabinoids, which included a feeling of “high”, drowsiness, somnolence, dysphoria, depression, paranoia, and hallucinations, caused some patients to withdraw from using cannabis as an antiemetic [346].

Another systematic review that included 28 studies (1772 participants) for CINV assessed the effectiveness of dronabinol (14 studies), nabilone (3 studies), nabiximols (1 study), levonandratol (4 studies), and THC (6 studies). Additionally, two studies included an ondansetron combination with other antiemetics such as prochlorperazine. Eight studies involved placebo control, with three of these involving an active comparator, and twenty studies included an active comparator. All trials showed a higher benefit of cannabinoids than other antiemetics and placebo; however, only a few had statistical significance. In three trials, cannabinoids compared to placebo showed a complete absence of nausea and vomiting (47% vs. 20%) response [329]. In patients on methotrexate, symptoms were significantly improved [347]. However, the same research group noted no effect of cannabis on patients receiving cyclophosphamide or doxorubicin after adding dronabinol [348]. Nabiximols, such as Sativex, were tested in 20 patients during a randomized crossover trial, in which 5 noted antiemetic effects [349].

The paucity of the existing clinical data, insufficient understanding of molecular mechanisms of ECS in CINV, and safety and efficacy of using cannabinoids in cancer patients justify the substantial need for more preclinical and clinical trials regarding cannabinoids.

#### 6.2.3. Improving Appetite

Cachexia and anorexia are one of the most troubling cancer-related symptoms experienced by patients. Cachexia is a multifactorial syndrome with progressive loss of skeletal muscular mass that cannot be reversed with the standard nutrition. A randomized placebo-controlled clinical trial conducted on 243 patients with cancer cachexia–anorexia showed no changes in appetite or quality of life under cannabinoids ingestion [145]. Three trials studied the effects of THC on appetite, food appreciation, calorie intake, and weight loss in patients with advanced cancers. As a result, in each study, an administration of oral THC improved one or more of the tested symptoms [297,298,350,351]. Another study with 469 cancer patients receiving dronabinol, megestrol, or both showed no advantage of THC alone or in combination with megestrol [352]. The findings suggest that cannabinoids are not as effective in cancer patients as they are in healthy subjects in terms of their appetite-stimulating effects. However, the palliative options for patients with advanced cancers are very limited, and cannabinoids merit further study in this context.

#### 6.2.4. Reducing Anxiety and Improving Sleep

A small parallel-group trial reported that CBD was associated with greater improvement in the anxiety factor compared with placebo during a simulated public speaking test in patients with a generalized anxiety disorder [329]. Four placebo-controlled studies in patients with chronic pain showed a greater benefit of dronabinol, nabilone, and nabiximols in reducing anxiety [329]. Another placebo-controlled study noted improved sleeping patterns in cancer patients with disordered chemosensory perception using dronabinol [297]. Additionally, a Canadian study assessing the quality of life in 74 patients with a newly diagnosed head and neck cancer reported that using marijuana significantly decreased anxiety/depression and pain/discomfort scores [304]. Currently, there is insufficient data regarding the effectiveness of cannabinoids for relieving the symptoms of anxiety in cancer patients. What is more, higher doses of cannabinoids may affect conventional anti-cancer treatment, which may limit the utility of cannabis as a palliative care.

## 7. Adverse, Unexpected, and Unintended Effects of Cannabinoids

There are also controversial data regarding the cannabinoid’s action on cancer cells. As stated in one published study, the administration of THC in a xenograft model of non-small cell lung carcinoma in immunodeficient mice showed antiangiogenic and antiproliferative effects [272]. However, some scientists reported that in immunocompetent animals, THC induced tumor growth, possibly due to its immunosuppressive effect [274,275]. On the other hand, the anti-inflammatory effects of endo- and phytocannabinoids can be used to prevent and treat colorectal cancer [353,354,355,356,357,358]. Such results are excellent proof that cannabinoids cannot be blindly taken as an anti-cancer agent in every case. Careful analysis of their various cellular effects, considering the molecular subtypes of cancer and possible drug interactions, must be done. Otherwise, they may cause more harm than benefit to struggling cancer patients.

One of the systematic reviews that evaluated 72 studies on cannabis showed that 62 studies reported adverse events (AEs) associated with cannabis use compared to controls. Cannabinoids were associated with a higher risk of AEs, serious AEs, and withdrawals due to AEs. The most common AEs included asthenia, balance problems, confusion, disorientation, diarrhea, euphoria, drowsiness, dry mouth, fatigue, hallucinations, nausea, somnolence, and vomiting [329]. However, there was no evidence as to whether the type of cannabinoid or mode of administration may affect the development of AEs [329]. These data suggest why cannabinoids are not the first line of treatment for various symptoms in cancer patients. If patients experience multiple AEs, they are likely to discontinue the medication and switch to something with the same or even lower potency, but with no AEs.

Overall, there is an extensive need for more well-designed, high-quality clinical trials regarding the anti-cancer and palliative properties of cannabinoids. However, as cannabis is classified as a Schedule I drug, it is difficult to conduct multicenter trials, regulatory hurdles delay such trials, and access to research-grade cannabis medications that match the products used by the cancer patients may be limited. All these factors affect cannabis research and data efficacy. Below, we summarize data from preclinical (Table 3) and clinical (Table 4) studies on the use of cannabinoids for tumor shrinkage and palliative care. As additional information, we compiled the list of clinical trials on cancer using cannabinoids sourced from clinicaltrials.gov as of 22 June 2022 in Appendix A.

## 8. Sex-Specific Differences in ECS, Ethical Considerations of Cannabis Use and Equal Access to Cannabis

Humans are diverse in numerous ways. We differ in sex, gender, sexuality, race, ethnicity, nationality, age, religious and cultural backgrounds, lifestyle, and more. Our different origins, life histories, preferences, and exposures shape who we are as individual human beings. As personalized medicine, also referred to as precision medicine, is developing and growing as a field, it becomes increasingly important to look at health and disease through the lens of diversity.

### 8.1. Sex-Specific Difference in Cancer and Use of Cannabis

Sex- and gender-based analysis is the first step toward proper implementation of precision medicine—a ground-breaking personalized approach aimed at tailoring disease diagnostics, treatment, and prevention to the needs of each patient based on genetics, epigenetics, environment, and the lifestyle of each individual. Sex encompasses biological attributes such as hormones, chromosomes, gene expression, anatomy, and physical features, while gender signifies the socially constructed behaviors, roles, expressions, and personalities of women, men, and gender-diverse people.

As cancer represents a leading cause of death globally, over the past few decades, a growing number of cancer epidemiology studies have reported the existence of sex disparities [359]. Significant sex disparities have been reported in cancer mortality, whereby lung, colorectal, esophageal, bladder, and stomach cancers, along with melanoma and leukemia, have higher mortality in males than in females [359,360,361]. Indeed, men overall have higher cancer incidence and mortality compared with women [359,362]. An exception is thyroid cancer, which occurs much more frequently in females than in males [363], while the incidence of colorectal, stomach, liver, and bladder cancers, as well as leukemia, is higher in males than in females [359,361,364]. An important and intriguing area of experimental and clinical oncology research is understanding the magnitude, nature, and mechanisms of sex differences in cancer predisposition, incidence, response to treatments, as well as mortality. Some of those may be associated with genetic and molecular changes, gene polymorphism in enzymes involved in drug metabolism, as well as functions of sex hormones that modulate gene expression in various cancers [359].

Treatment outcomes are also sex dependent, and genetic, molecular, and hormonal differences between males and females influence the effect of chemotherapy. While mounting evidence from preclinical models and clinical studies has reported sex disparities in chemotherapy outcomes, until now, chemotherapy has been administered without consideration of sex and gender differences, often leading to reduced efficacy, increased toxicity, and subpar outcomes [359].

Understanding sex and gender is critical in the field of medical cannabis. Indeed, sex differences exist in cannabis use, and research has begun to identify sex differences in the biological effects of cannabis. In recent years, the use of cannabis for pain relief has been consistently growing among women. Interestingly, sex differences have been reported in pain conditions and responses to pain medication, and new evidence is now suggesting that there may be sex differences in cannabinoid-mediated analgesia.

Emerging preclinical evidence, as well as data from human studies, strongly suggests sex differences in the endocannabinoid system (ECS) and cannabinoid pharmacology. Natural and synthetic cannabinoids led to sex differences in the antinociceptive response in animal models, which may correlate with those seen in the expression and function of ECS components. For example, female rodents were more sensitive to the effects of Δ9-tetrahydrocannabinol (Δ9-THC) due to the action of estradiol and progesterone, as well as differences in metabolism and cannabinoid receptor expression [365].

A recent rodent study reported the existence of sex differences in the ECS in two important regions of the central nervous system relevant to cortical spreading depression (V1M cortex) and descending modulatory networks in pain/anxiety (periaqueductal gray (PAG)). Analysis revealed significant differences in the concentrations of endocannabinoids 2-arachidonoylglycerol (2-AG) and anandamide between males and females, and these also varied between female estrous cycle stages. The 2-AG concentration was lower within the female PAG compared with the male PAG; this was confirmed using immunohistochemistry and proteomics. The observed sex differences in endogenous endocannabinoid mechanisms may in turn underlie the development of chronic pain conditions, as well as variations seen in therapy responses [366]. Sex differences in tolerance to THC were shown in mice with cisplatin-evoked chronic neuropathic pain [367]. Chronic adolescent exposure to cannabis in mice also resulted in sex-based changes in gene expression networks in the brain [368]. Interestingly, another rodent study reported that cannabinoids may be equally effective in both males and females in treating nausea [369].

Several other studies reported sex-driven modulation of endogenous cannabinoid signaling during the stress response [370], supporting the hypothesis that the ECS is engaged to a greater degree in males than in females during acute and severe stress and trauma, [371]. Indeed, the ECS plays a pivotal role in the activation and regulation of the stress response [372], albeit stress and anxiety disorders are more prevalent in women than in men [373].

The balanced function of the ECS is pivotal for maintaining mental health [374], and dysregulated endocannabinoid levels have been reported in humans with post-traumatic stress disorder [375]. Sex differences in the ECS function may underlie the sex differences regarding response to trauma and stress. Mechanistically, these differences may be due to hormonal regulation of endocannabinoids, where various sex differences are observed in production and function.

Furthermore, sex differences in the effects of cannabis may be attributed, at least in part, to the differences in fat-tissue distribution and muscle mass between males and females. Cannabinoids and other cannabis components are fat soluble and stored in fat cells, and women usually have higher percentages of body fat than men. Hence, women may experience different manifestations and magnitudes of cannabis effects.

Clinical trials that control for sex and stratify data by sex are pivotal for defining the extent to which medical cannabis and its components will be effective for both male and female patients. Moreover, researchers need to develop additional in-depth preclinical studies to uncover the impacts of sex in ECS functioning in health and disease [376].

While many studies included males and females to analyze palliative effects of cannabinoids on cancer [301,377,378,379], data were not analyzed as a function of sex. Two randomized, controlled trials enrolling healthy subjects analyzed sex differences in the acute effects of cannabis. The data showed that there were no variations in the acute effects of a moderate dose of vaporized cannabis between males and females [380]. A recent report by Meiri and coworkers compared medical-cannabis-related adverse effects between male and female patients with chronic non-cancer pain, and showed that sex differences exist, with adverse effects more frequently reported in females [381].

The only study focused on sex differences in the effects of cannabis on cancer was the large-scale analysis of the Quebec Cannabis Registry. Within the scope of this study, patients (171 males and 187 females) completed the revised Edmonton Symptom Assessment System (ESAS-r) questionnaire at baseline and three-month follow-up. The ESAS-r was used to assess pain, tiredness, anxiety, nausea, drowsiness, appetite, shortness of breath, and overall well-being. In addition, the interaction between sex and time on each ESAS-r symptom, as well as the interaction between time and THC:CBD ratios for each sex on total symptom burden, were analyzed [382]. While no sex differences were seen in the baseline ESAS-r scores, medical cannabis therapy led to significant improvements in pain, tiredness, anxiety, and well-being in both males and females. Improvements in drowsiness, nausea, appetite, and shortness of breath were seen only in females. Moreover, there were sex differences in the effects of THC-dominant cannabis, whereby it reduced pain only in males, and decreased nausea and led to overall improved well-being in females. This important pioneer study concluded that medical-cannabis-induced relief of cancer symptoms differs between sexes [382].

The majority of studies have analyzed cannabis as an adjunct modality used to control pain, anxiety, and treatment side effects in cancer patients. A recent Phase 1b study evaluated the safety and tolerability, as well as the pharmacokinetics and preliminary efficacy, of nabiximols and dose-intense temozolomide in 27 patients with recurrent glioblastoma following radiotherapy and temozolomide as first-line treatment [383]. The study enrolled males and females and laid the foundation for the future analysis of the potential efficacy of cannabinoids in recurrent glioblastoma. The large-scale study will help to establish the role of sex, if any, in the anti-cancer potential of cannabinoids.

### 8.2. Effect of Cannabis as a Function of Age

The effects of cannabis have to be analyzed across the age continuum. The use of cannabis is more common in adults, but it is expanding in pediatric populations. As yet, there are not enough data regarding the safety and efficacy of cannabis used as an anticancer agent or for symptom management in pediatric oncology [384]. Because the ECS system develops early in life, and in utero exposure data show negative outcomes, extreme caution is recommended in the use of cannabis in children and adolescents. Nevertheless, numerous studies and cases show the safety and efficacy of cannabis in the treatment of pediatric epilepsy and chemotherapy-induced nausea and vomiting (reviewed in [385]). A recent survey-based study of the use of cannabis in pediatric oncology showed that out of 14 participants who reported the use of a cannabis oil formulation for either cancer treatment or symptom management, all experienced symptom improvement [384]. Accumulating preclinical evidence suggests that cannabis may in turn have antitumor effects and may be a promising agent in pediatric oncology. Still, its use in pediatric practice remains controversial and requires more research [385].

More studies are needed to discern the safety and efficacy of cannabis for pain and cancer symptom management in older adults, as no studies have focused on this specific patient population or have stratified data by age, even though the majority of studies enrolled older adults. Cannabis has been shown to be effective in the palliative setting. A study of 2970 cancer patients (average age 59.5 ± 16.3 years) who received medical cannabis as palliative treatment for cancer showed that cannabis was a well-tolerated, effective, and safe option that helped patients cope with cancer-related symptoms such as pain, anxiety, and depression [291]. Analysis of cannabis effects in older adults is very important, as cancer is an aging-associated disease, and the prevalence of cannabis use among older adults (age 65 and older) for medicinal purposes is on the rise [386,387].

### 8.3. Equal Access to Cannabis for Everyone

Another important aspect of medical cannabis-based therapy is access, as cannabis-based therapies are still not covered by many major insurance plans and remain rather costly. Hence, cannabis-based formulations that may be used to manage cancer- and cancer treatment-related symptoms such as pain, mental health issues, loss of appetite, and cachexia may not be available to low-income and underprivileged groups of individuals, leading to even larger disparities in outcomes. Furthermore, there still is a huge stigma associated with cannabis use, which is often perceived as one of the bases of discrimination, alienation, and devaluation [388,389,390]. Up to 76% of patients who participated in the online cross-sectional survey study reported hiding their cannabis use from medical practitioners to avoid being judged [391]. Even in this day and age, a considerable number of approved medical cannabis patients in Canada report a lack of support and a need to conceal their medical cannabis use [391].

Stigmatization of medical cannabis users may also be more frequent in groups that already experience discrimination based on race and ethnic origins, sexual orientation, gender and gender identity, physical and mental disability, addiction, income, and age [392,393], causing even more trauma and distress that can lead to worsening health outcomes [394]. Regrettably, the medical establishment sometimes regards individuals experiencing poverty, mental health patients, and gender and sexual minorities as “problem patients,” and cannabis use may further contribute to their social devaluation and stigmatization [389,390].

Perceived stigmatization and discrimination and fear of judgment and rejection may sometimes lead to cancer patients refusing safe and effective cannabis-based adjuvant treatments, thus adding depression, anxiety, pain, suffering, stress, and trauma to an already grave and often deteriorating disease condition [390,395,396]. In an excellent review of stigma in medical cannabis use, Reid quoted a study by Rudski [396]: “Medicine can only be effective if it is taken, and stigma and lack of acceptability can interfere with compliance and safe access.” This is especially important in clinical oncology, where the benefits of adjuvant use of medical cannabis have been documented. Alienation, devaluation, and stigmatization of cannabis users may also depend upon their cultural, ethnic, and religious background, being more frequent in some communities and countries where cannabis use is still prohibited and criminalized [390]. As more studies are completed and results reported, the acceptance of medical cannabis will grow globally, leading to increased safe use and decreased stigma.

## 9. Conclusions

Cannabis and cannabinoids hold big promise for cancer therapy. First, however, we need to understand more about the role of ECS in normal human physiology and malignant transformations. How is it that all three components of ECS are typically upregulated in cancers, but addition of cannabinoids, overexpression, or sometimes downregulation of CB1/CB2 receptors actually inhibits the cancer growth? One hypothesis to be tested is that ECS is upregulated in cancer to a greater capacity to cope with the demands of continuous growth and that any changes in the balanced action of ECS—whether up- or downregulation—result in the growth inhibition or death of these cells.

Molecular mechanisms of ECS regulation and anti-cancer properties of cannabis also need to be clarified. What role do CB1 and CB2 receptors play? How other receptors contribute? How is it that some anti-cancer properties are independent of receptor activation?

Here, we showed substantial preclinical and clinical evidence of the potential of cannabinoids and cannabis extracts in primary and palliative care of cancer. However, we need more data on human consumption of cannabis, from case reports to clinical trials. As with any novel drug, we need to have good understanding about the interaction of individual active ingredients of the extracts with respect to their efficiency for primary and palliative care. Moreover, we need to know how cannabinoids interact with these drugs. We also actually need to find out what the active ingredients in cannabis are with respect to specific anti-cancer properties. Finally, we need to understand the sex-, gender- and age-specific differences in response to cannabis and cannabinoids.

## Figures and Tables

**Figure 1 cancers-14-05142-f001:**
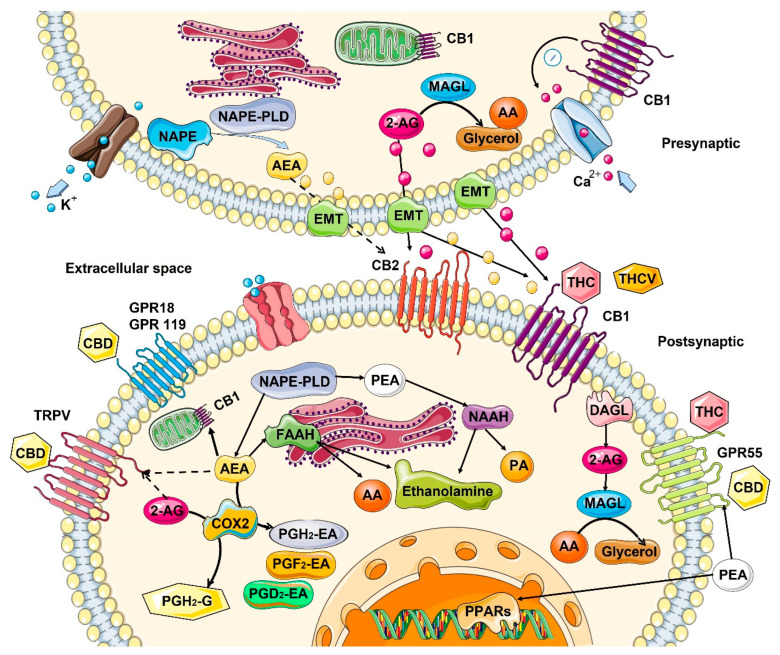
Biosynthesis, degradation, and interaction of endocannabinoids with cannabinoid receptors. Biosynthesis and the inactivation of the two endogenous lipid messengers, such as endocannabinoids N-Arachidonoylethanolamine or anandamide (AEA), 2-arachidonoylglycerol (2-AG), and N-palmitoylethanolamide (PEA) act on cannabinoid receptors. AEA and 2-AG are typically released on demand from membrane lipids. AEA synthesized from N-arachidonoyl-phosphatidylethanolamines (NAPE) via the activity of N-acyl-phosphatidylethanolamine-hydrolyzing phospholipase D (NAPE-PLD) and hydrolyzed by fatty acid amide hydrolase (FAAH) to ethanolamine and arachidonic acid (AA). 2-AG is 2-AG can also be produced from sn-2-arachidonate-containing diacylglycerols by sn-1-acyl-2-arachidonoylglycerol lipase (DAGL), and degraded by lipase (MAGL), releasing glycerol and AA. PEA is hydrolyzed by N-acylethanolamine-hydrolyzing acid amidase (NAAA) into ethanolamine and palmitic acid (PA). Cyclooxygenase-2 (COX-2) can also oxidize anandamide and 2-AG, followed by prostaglandin synthases to produce prostamides (from anandamide) and prostaglandin-ethanolamide, PG-EA (from 2-AG). Both AEA and 2-AG move across the plasma membrane via a purported endocannabinoid membrane transporter (EMT) and target CB1 and CB2, which show an extracellular binding site. 2-AG, AEA, and PEA directly activate orphan G-protein-coupled receptors (GPR55, GPR18, GPR119), the transient receptor potential of vanilloid (TRPV) channel, and peroxisome proliferator-activated nuclear receptors (PPARs). Dashed lines denote low-affinity bindings. Phytocannabinoids Δ9-tetrahydrocannabinol (THC), cannabidiol (CBD), and Δ9-tetrahydrocannabivarin (THCV) showed to activate cannabinoid receptors. CB1, cannabinoid receptor 1; CB2, cannabinoid receptor 2; ER, endoplasmic reticulum. This figure was created using images from Servier Medical Art Commons Attribution 3.0 Unported License (http://smart.servier.com (accessed on 4 May 2022)).

**Figure 2 cancers-14-05142-f002:**
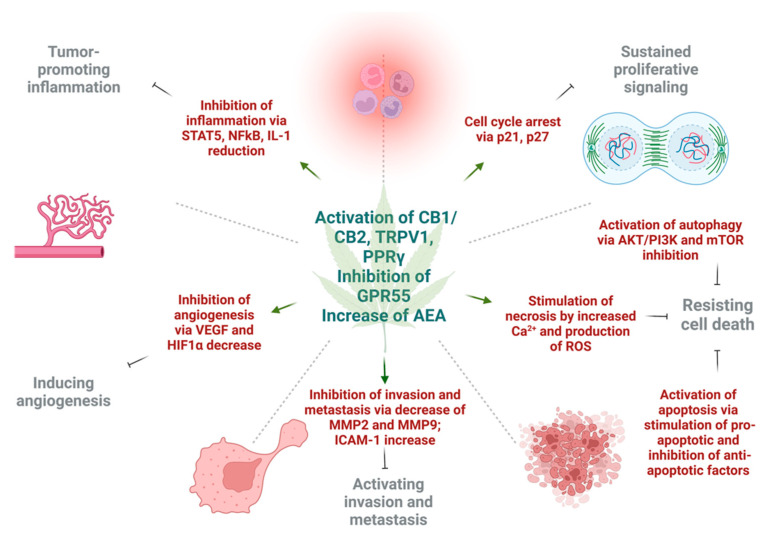
The effects of cannabinoids on different hallmarks of cancer. The activation of cannabinoid 1 (CB1), cannabinoid 2 (CB2), and transient receptor potential cation channel 1 (TRPV1) increase levels of anandamide (AEA), as well as inhibition of G-protein coupled receptor 55 (GPR55), and exert different effects on tumor cells in respect to cancer hallmarks. Cannabinoids inhibit tumor-promoting inflammation via downregulation of nuclear factor κB (NFκB), signal transducer and activator of transcription 5 (STAT5), and interleukin 1 (IL1). The angiogenesis is inhibited by reduction in vascular endothelial growth factor (VEGF) and hypoxia inducible factor 1α (HIF1α). Next, invasion and metastasis are prevented by decrease of tissue degrading enzymes—matrix metalloproteinase 2 and 9 (MMP2, MMP9), as well as expression of intercellular adhesion molecule 1 (ICAM-1). Sustained proliferative signaling is opposed by the activation of p21 and p27 that leads to cell cycle arrest. Lastly, under the action of cannabinoids, cell death may be achieved by three mechanisms. Autophagy is triggered by inhibition of protein kinase B (AKT), phosphoinositide 3-kinase (PI3K) and mammalian inhibitor of rapamycin (mTOR). Apoptotic cell death is a result of upregulation of pro-apoptotic and downregulation of anti-apoptotic factors under the action of different cannabinoids. Necrosis can result due to high Ca^2+^ release and formation of ROS in cancer cells. This figure was created with BioRender.com.

**Table 1 cancers-14-05142-t001:** Age-dependent changes to ECS components in different normal tissues.

Tissues/Organs	Endocannabinoids	Receptors	Metabolizing Enzymes
Skin	No reliable data	↓ in CB1 expression [13]	*FAAH* tends to ↓ with age [45]
Lung	2-AG ↓ and AEA ↑ in mice [46]	No reliable data	No reliable data
Brain	From no change [40] to a ↓ in AEA [47] ↓ in 2-AG levels in mice [48]	From ↑ in humans [41] to no change [39] to a ↓ [38,49] in mice/rats in CB1 expression, brain area-specific	↓ *FAAH* activity in rats [50] ↑ in MAGL levels in mice [48]
Blood	Small ↑ in 2-AG and AEA in mice [46]	No reliable data	No reliable data

↑ Indicates increased expression or the amount of circulating product, while ↓ indicates decreased amounts.

**Table 2 cancers-14-05142-t002:** Changes to ECS components in malignant tissues.

Tissues/Organs	Endocannabinoids	Receptors	Metabolizing Enzymes
Skin	Decreased AEA and increased 2-AG in melanoma [51]	↑ CB2 in melanoma [52]	↑ MAGL [53], and ↑ FAAH in melanoma [45]
Intestine (colorectal)	↑ AEA and 2-AG [54,55] ↑ LPI [55,56]	↓ CB1 [57,58], ↑ GPR55 [57,59], ↑ CB2 and ↓ CB1 [58]	↑ MAGL [54], ↑ *FAAH* [45]
Lung		↑ CB1 and CB2 [60,61]	↑ FAAH [62], ↑ *FAAH* [45]
Breast	↑ LPI [63]	↑ CB1 and CB2 [64,65,66,67], ↑ GPR55 [68,69]	↑ *FAAH* [45]
Brain	↑ AEA and 2-AG in many cancers [70,71,72,73,74]	↑ *CNR1*, ↑ *CNR2*, ↑ CB1 and CB2 in glioma [75,76,77,78], ↓ CB1 in glioma [79]	↓ FAAH [80] in glioma, ↑ *FAAH* in glioma [45]

Changes in ECS components in major cancers. ↑ indicates upregulation, while ↓ indicates downregulation. Italic indicates gene expression (*CNR1*/*CNR2*, for example), while non-italic—proteins (CB1/CB2, for example). LPI—lysophosphatidylinositol.

**Table 3 cancers-14-05142-t003:** Preclinical studies regarding the primary (tumor shrinkage) and palliative effects of cannabinoids on different types of cancer.

Cannabis Drugs Used	Cancer Types/Preclinical Models of Diseases	Experimental Models	Effects in Cancer	Citation
Delta-9-THC	Hepatic adenomas, hepatocellular carcinoma, decreased incidence in adenomas and papillomas in mammary glands, uterus, pituitary gland, testicles, pancreas	In vivo, in vitro	Cancer prevention	[191]
Delta-9-THC, delta-8-THC, Selective CB2 agonist JWH-133, Co-administration of CBD and THC	Lewis lung adenocarcinoma, glioblastoma multiforme	In vivo, in vitro	Reduced tumor growth	[70,75,144,145]
Delta-9-THC, HU-210, anandamide, CB2 agonist JWH-015	Malignant lymphoblastic diseases	In vivo, in vitro	Reduced tumor growth	[271]
CB1/CB2 agonist WIN-55,212-2, CB2 agonist JWH-133	Non-melanoma skin tumors	In vivo, in vitro	Reduced tumor growth	[92]
Delta-9-THC, CB2 agonist JWH-015	Hepatocellular carcinoma	In vitro, in vivo	Reduced tumor growth	[195]
Delta-9-THC, WIN55,212-2, JWH-015	Non-small cell lung carcinoma	In vitro, in vivo (immunodeficient mice)	Reduced tumor growth	[61,272]
CBD, THC, JWH-015	Breast cancer	In vitro, in vivo	Reduced tumor growth	[67,151,273]
CBD	Colorectal cancer	In vitro, in vivo	Reduced tumor growth	[218,219]
Delta-9-THC	Non-small cell lung carcinoma, breast cancer	In vitro, in vivo (immunocompetent mice)	Increased tumor growth	[274,275]
Delta-9-THC, CP-55,940, WIN55,212-2, CBD	Animal model of emesis	In vivo	Antiemetic effect, inhibition of chemotherapy-induced nausea and vomiting	[276,277,278]
Anandamide, Delta-9-THC	Changes in appetite in animals	In vivo	Increased food intake	[279,280]
WIN55,212-2, arachidonylcyclopropylamide, AM1241	Animal models of pain induction	In vivo	Analgetic effect	[281,282,283]
CBD, WIN55,212-2	Animal models of chemotherapy-induced neuropathic pain	In vivo	Analgetic effect	[284,285]
CBD	Animal models of stress, recording sleep-walking cycles in rats	In vivo	Reduction of anxiety and improvement of sleep	[286,287]

**Table 4 cancers-14-05142-t004:** Clinical studies regarding the primary (anti-cancer), palliative, and adverse effects of cannabinoids and cannabis use in cancer patients.

Drugs Used	Cancer Types/Participant Groups	Primary/Anticancer Effects	Palliative Care	Adverse Effects	Citation/Clinical Trial #
Delta-9-THC	Intra-tumoral injection in patients with recurrent glioblastoma multiforme	No significant clinical benefit	-	-	[200,288]
Sativex and Temozolomide	Glioblastoma multiforme	Increased 1-year survival rate in 39%	-	-	NCT01812603; NCT01812616 [289]
Dexanabinol	Solid tumors	Progression-free survival increased	-	-	NCT01489826
CBD	Acute leukemia and myelodysplastic syndrome	Lower incidence rate of acute graft-versus-host disease after allogenic hematopoietic stem cell transplantation	-	-	NCT01385124 NCT01596075 [290]
Government-issued Cannabis	Cancer patients	-	Improvement of symptoms related to nausea and vomiting, sleep disorders, restlessness, anxiety and depression, pruritus, headaches	-	[291]
Cannabis use	Breast cancer patients	-	Relief symptoms: of pain, insomnia, anxiety, stress, nausea, and vomiting	-	[292]
Dronabinol, Nabilone *	Patients with different cancers	-	Treatment of chemotherapy-induced nausea and vomiting	-	[293,294,295,296]
Delta-9-THC	Patients with advanced cancers	-	Appetite stimulation	-	[296,297,298]
Delta-9-THC, THC:CBD extracts, Nabilone	Patients with different cancers	-	Analgetic effect	-	[299,300,301,302]
Cannabis	Patients with chemotherapy-induced peripheral neuropathy	-	Analgetic effect	-	[303]
Delta-9-THC, Dronabinol *	Patients with different cancers	-	Anxiolytic effect, increased quality of sleep	-	[296,302]
Cannabis use	Patients with head and neck cancers	-	Decreased anxiety and depression scores	-	[304]
Marijuana	Chronic marijuana smokers	-	-	Increased risk for testicular germ cell tumors in “Heavy” Cannabis users	[305]
Smoking Cannabis	Healthy subjects and Cannabis users	-	-	Cannabis use was associated with 45% reduction in bladder cancer incidence	[306]
Smoking Cannabis	Healthy individuals, lung cancer patients	-	-	Smoking cannabis is not associated with lung cancer or head and neck cancers	[307,308,309]
Cannabis use during nivolumab immunotherapy	Patients with advanced melanoma, non-small cell lung carcinoma, renal cell carcinoma	-	-	Cannabis use reduced the response rate to immunotherapy by 21.25%. Cannabis use was correlated with poor clinical outcome	[310,311]

* Dronabinol and nabilone are currently approved for treatment of cancer-related side effects. # Numbers of clinical trials related to cannabinoid use in cancer (source clinicaltrials.gov as of 22 June 2022).

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
