# Peer review of "Use of Cannabis and Cannabinoids for Treatment of Cancer"

_cancers, 2022, doi:10.3390/cancers14205142_

Round 1

Reviewer 1 Report

The review by Cherkasova et al entitled   "Use of cannabis and cannabinoids for treatment of cancer " is a well-written review  on cannabis and cannabinoids hypothetical use  for cancer therapy. The review is a comprehensive and precise description of reported mechanisms regulating  cannabinoid effects   , their receptors and regulators of their metabolism. On the other hand , also considering the enormous body of findings reported, not always consistent each other, and most of conclusions drawn in the paper,  showing no certainty about the possible effects of cannabinoids on normal and malignant cells,  (for instance, as reported in the Conclusion session by authors : " First, however, we need to understand more about the role of ECS in normal human physiology and malignant transformations. How is it that all three components of ECS are typically upregulated in cancers, but addition of cannabinoids, overexpression or sometimes downregulation of CB1/CB2 receptors actually inhibits the cancer growth?"). this reviewer believe that this filed is too "young" to draw any conclusion on its potential role in cancer. For these reasons,  I judge this paper not suitable for publication in  Cancers.

Author Response

We would like to thank the reviewer for their thoughts on our review. We have removed some ambiguity in our conclusions and future perspectives in this field. There is plenty of good science done and plenty of results published demonstrating the positive effect of cannabinoids and cannabis extracts on various aspects of cancer. At the same time, caution needs to be exercised as the effects were cancer-, tissue- and compound-specific. Also, as we have summarized, the mechanisms of anti-cancerous activity of cannabis are not yet clear, and the role of ECS system is not fully understood. There is nothing wrong about that. We believe we have provided a balanced view on current cannabis/cancer science, and this is why we think that our review deserves to be published.

Reviewer 2 Report

The Manuscript is an extensive review on the use of cannabis and cannabinoids for treatment of cancer. The authors clearly describe many aspects of the advantages and disadvantages of this therapy. To improve the paper, they could add a table showing the drugs used/tested and their correlation with pre-clinical and clinical aspects of therapy, including data on subjects/animals, responses or adverse effects.

Author Response

We would like to thank the reviewer for this suggestion. In this submission, we have introduced two tables summarizing current data on the use of cannabis/cannabinoids drugs on cancers in animal/human studies.

Reviewer 3 Report

The review is well-balanced and comprehensive. The authors summarized the knowledge of cannabis and cannabinoids for cancer treatment; the content is potentially interesting and attractive for future investigation. In addition, the paper might provide good incentives and motivation for the field.
Minor point
The gender section on the differences in the ECS is overly long and repetitive in some places.

Author Response

We agree with the reviewer, and we have shortened that part of the manuscript, removing any duplications/redundancy.